# Why do wide stochastic neural networks have vanishing variance?

## Abstract

This work studies the prediction variance of stochastic neural networks, a main type of neural network in use. We constructively prove that as the width of an optimized stochastic neural network tends to infinity, its predictive variance on the training set decreases to zero. In particular, we show that a solution with vanishing variance exists when the model has a "two-layer" structure, where the upper layer can copy independent copies of the latent variable, and the second layer can average over such copies to cancel the noise. The main implication of our result is that the popular belief that a powerful decoder causes the neural network prediction variance to vanish is not the full picture. Two common examples of learning systems that our theory can be relevant to are neural networks with dropout and Bayesian latent variable models in a special limit. Our result thus helps us better understand how stochasticity affects the learning of neural networks.

## 1 Introduction

Applications of neural networks have achieved great success in various fields. A major extension of the standard neural networks is to make them stochastic, namely, to make the output a random function of the input (Ajay et al., 2018; Su et al., 2019; Raissi, 2018; Kwon et al., 2020). In a broad sense, stochastic neural networks include neural networks trained with dropout (Srivastava et al., 2014; Gal & Ghahramani, 2016), Bayesian networks (Mackay, 1992), variational autoencoders (VAE) (Kingma & Welling, 2013), and generative adversarial networks (Goodfellow et al., 2014). In this work, we formulate a rather broad definition of a stochastic neural network in Section 3. There are many reasons why one wants to make a neural network stochastic. Two main reasons are (1) regularization and (2) distribution modeling. Since neural networks with stochastic latent layers are more difficult to train, stochasticity is sometimes believed to help regularize the model and prevent memorization of samples (Srivastava et al., 2014). The second reason is easier to understand from the perspective of latent variable models. By making the network stochastic, one implicitly assumes that there exist latent random variables that generate the data through some unknown function. Therefore, by sampling these latent variables, we are performing a Monte Carlo sampling from the underlying data distribution, which allows us to model the underlying data distribution by a neural network. This type of logic is often invoked to motivate the VAE and GAN. Therefore, stochastic networks are of both practical and theoretical importance to study. In the related fields, a mainstream belief about a stochastic neural network is that it overfits any distribution easily because neural networks can approximate many sufficiently regular distribution arbitrarily well (Lee et al., 2017); when this happens, the model ignores the stochastic components of the architecture and outputs with very low variance. In Bayesian deep learning, the strong power of the decoder has been conjectured to lead to various pathological phenomena in deep learning, such as posterior collapse (Alemi et al., 2018), where the model output has a vanishing variation. However, there is no direct proof of whether high expressivity or strong power is the cause of such problems and how a neural network can actually reach such a pathological state. Our work answers this question of how a neural network can achieve such a state directly and fills in this important theoretical gap.

In this work, we theoretically study stochastic neural networks. We prove with an explicit construction that as the width of an optimized stochastic net increases to infinity, its predictive variance decreases to zero on the training set. See Figure 1 for an illustration of this effect. Along with this proof, we propose a novel

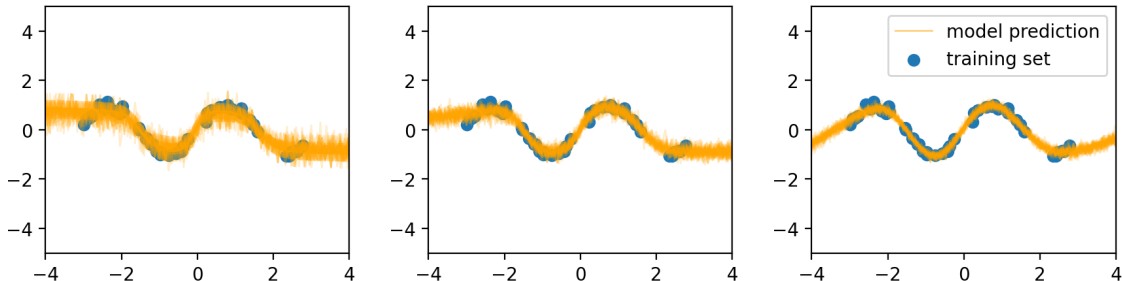

Figure 1: Distribution of the prediction of a trained neural network with dropout. We see that as the hidden width $d$ increases, the spread of the prediction decreases. **Left**: $d = 10$. **Mid**: $d = 50$, **Right**: $d = 500$. See Section 4.1 for a detailed description of the experimental setup.

theoretical framework that allows us to abstract away the specific definitions of contemporary architectures of neural networks and makes our result applicable to a family of functions that includes many common neural networks as a strict subset.

This work is organized as follows. We discuss the related works in the next section. Section 3 presents the main theoretical contributions and their implications. Section 4 substantiates the theory with numerical simulations. Appendix Section A presents some additional experiments. Appendix Section B presents all the proofs and some additional theoretical results.

## 2 Related Works

Bayesian inference promises to model the statistics of $y$. For example, we would like stochastic neural networks to have well-calibrated uncertainty estimates, a trait that is highly desirable for practical, safe, and reliable applications (Wilson & Izmailov, 2020; Gawlikowski et al., 2021; Izmailov et al., 2021). This expectation means that a well-trained stochastic network should have a predictive variance that matches the actual level of randomness in the labeling. Two applications we consider in this work are dropout (Srivastava et al., 2014), which can be interpreted as a stochastic technique for approximate Bayesian inference, and VAE (Kingma & Welling, 2013), which is among the main Bayesian deep learning methods in use. Theoretically, while a unified approach is lacking, some previous works exist to separately study different stochastic techniques in deep learning. A series of recent works approaches the VAE loss theoretically (Dai & Wipf, 2019). Another line of recent works analyzes linear models trained with VAE to study the commonly encountered mode collapse problem of VAE (Lucas et al., 2019; Koehler et al., 2021). In the case of dropout, Gal & Ghahramani (2016) establishes the connection between the dropout technique and Bayesian learning. Another series of work extensively studied the dropout technique with a linear network (Cavazza et al., 2018; Mianjy & Arora, 2019; Arora et al., 2020) and showed that dropout effectively controls the rank of the learned solution and approximates a data-dependent $L_2$ regularization. In the literature of Bayesian neural networks (BNN), Farquhar et al. (2020) showed that a sufficiently wide mean-field BNN could approximate any sufficiently regular distribution. Foong et al. (2020) studies the in-between predictive variance of deep nonlinear networks.

In Bayesian deep learning, a line of works suggested that a predominant problem in training with ELBO is that the decoder is often too powerful (Bowman et al., 2015; Chen et al., 2016; Alemi et al., 2018). A work that is particularly relevant to our works is Alemi et al. (2018), which showed that when the decoder is sufficiently powerful to model the empirical data distribution, the model will learn to ignore the latent variables $z$ and perfectly reconstruct the data distribution $\hat{p}(x)$. When the empirical data distribution is a mixture of delta distributions, this work implies that a sufficiently powerful decoder will have a vanishing predictive variance. However, it is unclear from these works what it means to be "powerful" for a neural network or how it could be achieved. Our work significantly advances this conventional perspective in two ways. First of all, we make a positive construction of a "powerful" neural network that has a vanishing variance. Second, we show a powerful decoder is not required for a vanishing variance (or for ignoring the latent variable $z$). For example, having a powerful encoder can also make the model "ignore" the latent variable and leads to a vanishing variance.

# 3 Main Result

In this section, we present and discuss our main result. Notation-wise, let $W$ denote a matrix, $W_{j:}$ denote its $j$-th row viewed as a vector. Let $v$ be any vector, and $v_j$ denote its $j$-th element; however, when $v$ involves a complicated expression, we denote its $j$-th element as $[v]_j$ for clarity.

## 3.1 Problem Setting

We first introduce two basic assumptions of the network structure.

**Assumption 1.** (Neural networks can be decomposed into Lipshitz-continuous blocks.) Let $f$ be a neural network. We assume that there exist functions $g^1$ and $g^2$ such that $f = g^2 \circ g^1$, where $\circ$ denotes functional composition. Additionally, both $g^1$ and $g^2$ are Lipshitz-continuous.

Throughout this work, the component functions $g^1$, $g^2$ of a network $f$ are called a block, which can be seen as a generalization of a layer. It is appropriate to call $g^1$ the input block and $g^2$ the output block. Because the Lipshitz constant of a neural network can be upper bounded by the product of the largest eigenvalue of each weight matrix times the Lipshitz constant of the non-linearity, the assumption that every block is Lipshitz-continuous applies to all existing networks with fixed weights and with Lipshitz-continuous activation functions (such as ReLU, tanh, Swish (Ramachandran et al., 2017), Snake (Ziyin et al., 2020) etc.).

If we restrict ourselves to feedforward architectures, we can discuss the meaning of an "increasing width" without much ambiguity. However, in our work, since the definition of blocks (layers) is abstract, it is not immediately clear what it means to "increase the width of a block." The following definition makes it clear that one needs to specify a sequence of blocks to define an increasing width.[1]

**Definition 1.** (Models with an increasing width.) Each block of a neural network $f$ is labeled with two indices $d_1, d_2 \in \mathbb{Z}^+$. Let $f = g^2 \circ g^1$; we write $g^i = {}^{d_2,d_1}g^i$ if for all $x$, ${}^{d_2,d_1}g^i(x) \in \mathbb{R}^{d_2}$ and $x \in \mathbb{R}^{d_1}$. Moreover, to every block $g$, there corresponds a countable set of blocks $\{{}^{i,j}g\}_{i,j\in\mathbb{Z}^+}$. For a block $g$, its corresponding block set is denoted as $\mathcal{S}(g) = \{{}^{i,j}g\}_{i,j\in\mathbb{Z}^+}$. Also, to every sequence of blocks $g^1, g^2, ...,$ there also corresponds a sequence of parameter sets $w^1, w^2, ...$ such that $g^i = g^i_{w^i}$ is parametrized by $w^i$. The corresponding parameter set of block $g$ is denoted as $w(g)$.

Note that if $f = g^2 \circ g^1$, $g^1 = {}^{d_2,d_1}g^1$ and $g^2 = {}^{d_4,d_3}g^2$, $d_2$ must be equal to $d_3$; namely, specifying $g^1$ constrains the input dimension of the next block. It is appropriate to call $d_2$ the width of the block ${}^{d_2,d_1}g$. Since each block is parametrized by its own parameter set, the union of all parameter sets is the parameter set of the neural network $f$: $f = f_w$. Since every block comes with the respective indices and equipped with its own parameter set, we omit specifying the indices and the parameter set when unnecessary. The next assumption specifies what it means to have a larger width.

**Assumption 2.** (A model with larger width can express a model with smaller width.) Let $g$ be a block and $\mathcal{S}(g)$ its block set. Each block $g = g_w$ in $\mathcal{S}(g)$ is associated with a set of parameters $w$ such that for any pair of functions ${}^{d_2,d_1}g, {}^{d_2',d_1}g' \in \mathcal{S}(g)$ such that $d_2' > d_2$, any fixed $w$, and any surjective mappings $m$ from $\{1,...,d_2'\} \to \{1,...,d_2\}$, there exists parameters $w'$ such that $[{}^{d_2,d_1}g_w(x)]_{m(l)} = [{}^{d_2',d_1}g_{w'}(x)]_l$ for all $x$ and $l$.

This assumption can be seen as a constraint on the types of block sets $\mathcal{S}(w)$ we can choose. As a concrete example, the following proposition shows that the block set induced by a linear layer with arbitrary input and output dimensions followed by an element-wise non-linearity satisfies our assumption.

**Proposition 1.** *Let ${}^{d_2,d_1}g_{W,b}(x) = \sigma(Wx + b)$ where $\sigma$ is an element-wise function, $W \in \mathbb{R}^{d_1 \times d_2}$, and $b \in \mathbb{R}^{d_2}$. Then, $\mathcal{S}(g) = \{{}^{i,j}g_{W,b}(x)\}_{i,j\in\mathbb{Z}^+}$ satisfies Assumption 2.*

*Proof.* Consider two functions ${}^{d_1,d_2}g_{\{W,b\}}(x)$ and ${}^{d_1',d_2}g_{\{W',b'\}}(x)$ in $\mathcal{S}(g)$. Let $m$ be an arbitrary mapping from $\{1,...,d_1'\} \to \{1,...,d_1\}$. It suffices to show that there exist $W'$ and $b'$ such that $[{}^{d_1,d_2}g_{\{W,b\}}(x)]_{m(l)} - [{}^{d_1',d_2}g_{\{W',b'\}}(x)]_l = 0$ for all $l$. For a matrix $M$, we use $M_{j:}$ to denote the $j$-th row of $M$. By definition, this

---

[1]Also, note that this definition of "width" makes it possible to define different ways of "increasing" the width and is thus more general than the standard procedure of simply increasing the output dimension of the corresponding linear transformation.

condition is equivalent to

$$\sigma(W_{m(l):}x + b) = \sigma(W'_{l:}x + b') \tag{1}$$

which is achieved by setting $b' = b$ and $W'_{l:} = W_{m(l):}$, where $W_{m(l):}$ is the $m(l)$-th row of $W$. $\square$

Now, we are ready to define a stochastic neural network.

**Definition 2.** (*Stochastic Neural Networks*) A neural network $f = g^2 \circ g^1$ is said to be a stochastic neural network with stochastic block $g^1$ if $g^1 = g^1(x, \epsilon)$ is a function of $x$ and a random vector $\epsilon$, and the corresponding deterministic function $f' := g^2 \circ h^1$ satisfies Assumption 2, where $h^1 = \mathbb{E}_\epsilon \circ g^1$.

Namely, a stochastic network becomes a proper neural network when averaged over the noise of the stochastic block. To proceed, we make the following assumption about the randomness in the stochastic layer. We note that this definition allows for multiple two-block decompositions of the same neural network. For example, For a two layer neural network with tanh activation, $f(x) = U \tanh(Wx)$, one can either define $W$ or $\tanh \circ W$ as $g^1$, and the rest of the model as $g^2$.

**Assumption 3.** (*Uncorrelated noise*) For a stochastic block $g$, $\mathrm{Cov}_\epsilon[g(x,\epsilon), g(x,\epsilon)] = \Sigma(x)$, where $\Sigma$ is a diagonal matrix and $\Sigma_{ii} < \infty$ for all $i$.[2]

This assumption applies to standard stochastic techniques in deep learning, such as dropout or the reparametrization trick used in approximate Bayesian deep learning. Lastly, we assume the following condition for the architecture.

**Assumption 4.** (*Stochastic block is followed by linear transformation.*) Let $f = g^2 \circ g^1$ be the stochastic neural network under consideration, and let $g^1$ be the stochastic layer. We assume that for all $^{i,j}g \in \mathcal{S}(g^2)$, $^{i,j}g_w = g'_{w'}(Wx + b)$ for a fixed function $g' : \mathbb{R}^d \to \mathbb{R}^i$ with parameter set $w'$, where $W \in \mathbb{R}^{d \times j}$ and bias $b \in \mathbb{R}^d$ for a fixed integer $d$. In our main result, we further assume that $b = 0$ for notational conciseness.

In other words, we assume that the second block $g^2$ can always be decomposed as $g' \circ M$, such that $M$ is an optimizable linear transformation. This is the only architectural assumption we make. In principle, this can be replaced by weaker assumptions. However, we leave this as an important future work because Assumption 4 is sufficient for the purpose of this work and is general enough for the applications we consider (such as dropout and VAE). We also stress that the condition that $g^2$ starts with a linear transformation does not mean that the first proper layer of $g^2$ is linear. Instead, $M$ can be followed by an arbitrary Lipshitz activation function as is usual in practice; in our definition, if it exists, this following activation is assumed into the definition of $g'$.

The actual rather restrictive assumption in Assumption 4 is that the function $g'$ has a fixed input dimension (like a "bottleneck"). In practice, when one scales up the model, it is often the case that one wants to scale up the width of all other layers simultaneously. For the first block, this is allowed by assumption 2.[3] We note that this bottleneck assumption is mainly for notational concision. In the appendix B.3, we show that one can also extend the result to the case when the input dimension (and the intermediate dimensions) of $g'$ also increases as one increases the width of the stochastic layer.

**Problem Setting Summary**. To summarize, we require a network $f$ to be decomposed into two blocks: $f = g^2 \circ g^1$ and $g^1$ is a stochastic block. Each block is associated with its indices, which specify its input and output dimensions, and a parameter set that we optimize over. For example, we can write a block $g$ as $g = {}^{d_2,d_1}g^i_w$ to specify that $g$ is the $i$-th block in a neural network, is a mapping from $\mathbb{R}^{d_1}$ to $\mathbb{R}^{d_2}$, and that its parameters are $w$. However, for notational conciseness and better readability, we omit some of the specifications when the context is clear. For the parameter $w$, sometimes, we view $w$ as a set and discuss its unions and subsets; for example, let $f_w = g^2_{w^2} \circ g^1_{w^1}$; then, we say that the parameter set $w$ of $f$ is the union of the parameter set of $g^1$ and $g^2$: $w = w^1 \cup w^2$. Alternatively, we also view $w$ as a vector in a subset of the real space, so that we can look for the minimizer $w$ in such a space (in expressions such as $\min_w L(w)$).

---

[2]Some notable examples that this assumption does not apply to are the full-covariance models such as the ones proposed in (Louizos & Welling, 2016).

[3]For example, if $g^1$ is a multilayer perceptron, it is easy to check that assumption 2 is satisfied if one increases the intermediate layers of $g^1$ simultaneously.

### 3.2 Convergence without Prior

In this work, we restrict our theoretical result to the MSE loss. Consider an arbitrary training set $\{(x_i, y_i)\}_{i=1}^N$, when training the deterministic network, we want to find

$$w_* = \arg\min_w \sum_{i=1}^N [f_w(x_i) - y_i]^2. \tag{2}$$

It is convenient to write $[f_w(x_i) - y_i]^2$ as $L_i(w)$. We note that it is well-known that the minimizer of the MSE should have a minimal variance, and the actual technical contribution of this section is to give explicit constructive proof of how such solutions can be achieved in a neural network.

An overparametrized network can be defined as a network that can achieve zero training loss on such a dataset.

**Definition 3.** A neural network $f_w$ is said to be overparametrized for a non-negative differentiable loss function $\sum_i L_i$ if there exists $w_*$ such that $\sum_i L_i(w_*) = 0$. For a stochastic neural network $f$ with stochastic block $g^1$, $f$ is said to be overparametrized if $f' := g^2 \circ \mathbb{E}_\epsilon \circ g^1$ is overparametrized, where $\mathbb{E}_\epsilon$ is the expectation operation that averages over $\epsilon$.

Namely, we say that a stochastic network is overparametrized if its deterministic part is overparametrized. When there is no degeneracy in the data (if $x_i = x_j$, then $y_i = y_j$), zero training loss can be achieved for a wide enough neural network, and this definition is essentially equivalent to assuming that there is no data degeneracy and is thus not a strong limitation of our result. A crucial remark is that this assumption does *not* imply that the corresponding stochastic network can express any distribution. For example, for linearly interpolable data, a two-layer linear network is overparametrized, but its expressivity of distributions does not increase with width at all. This point will become crucial when we discuss the implication of our results in Section 3.5.

With a stochastic block, the training loss becomes (due to the sampling of the hidden representation)

$$\mathbb{E}_\epsilon \left[ \sum_i^N L_i \right] = \sum_{i=1}^N \mathbb{E}_\epsilon \left[ (f_w(x_i, \epsilon) - y_i)^2 \right]. \tag{3}$$

Note that this loss function can still be reduced to 0 if $f_w(x_i, \epsilon) = y_i$ for all $i$ with probability 1, though this is in general not possible for a stochastic network to achieve.

With these definitions at hand, we are ready to state our main result.

**Theorem 1.** *Let the neural network under consideration satisfy Assumptions 1 ,2, 3 and 4, and assume that the loss function is given by equation 3. Let $\{^{d_1}f\}_{d_1 \in \mathbb{Z}^+}$ be a sequence of stochastic networks such that, for fixed integers $d_2, d_0$, $^{d_1}f = {}^{d_2,d_1}g^2 \circ {}^{d_1,d_0}g^1$ with stochastic block $^{d_1,d_0}g^1_{w(g_1)} \in \mathcal{S}(g^1)$. Let $^{d_1}f$ be overparameterized for all $d_1 \geq d^*$ for some $d^* > 0$. Let $w_* = \arg\min_w \sum_i^N \mathbb{E}[L(^{d_i}f_w(x, \epsilon), y_i)]$ be a global minimum of the loss function. Then, for all $x$ in the training set,*

$$\lim_{k \to \infty} Var_\epsilon \left[ {}^{kd^*}f_{w_*}(x, \epsilon) \right] = 0. \tag{4}$$

*Proof Sketch.* The full proof is given in Appendix Section B.1. In the proof, we denote the term $L(^{d_1}f_w(x, \epsilon), y_i)$ as $L_i^{d_1}(w)$. Let $w_*$ be the global minimizer of $\sum_j^N \mathbb{E}_\epsilon \left[ L_j^{d_1}(w) \right]$. Then, for any $w$, by definition of the global minimum,

$$0 \leq \sum_j^N \mathbb{E}_\epsilon \left[ L_j^{d_1}(w_*) \right] \leq \sum_j^N \mathbb{E}_\epsilon \left[ L_j^{d_1}(w) \right]. \tag{5}$$

If $\lim_{d_1 \to \infty} \sum_j^N \mathbb{E}_\epsilon \left[ L_j^{d_1}(w) \right] = 0$, we have $\lim_{d_1 \to \infty} \sum_j^N \mathbb{E}_\epsilon \left[ L_j^{d_1}(w_*) \right] = 0$, which implies that $\lim_{d_1 \to \infty} \mathbb{E}_\epsilon \left[ L_j^{d_1}(w_*) \right] = 0$ for all $j$. By bias-variance decomposition of the MSE, this, in turn, implies that $Var[^{d_1}f_{w_*}(x_j)] = 0$ for all $j$. Therefore, it is sufficient to construct a sequence of $w$ such that $\lim_{d_1 \to \infty} \mathbb{E}_\epsilon \sum_j^N \left[ L_j^{d_1}(w) \right] = 0$. The rest of the proof shows that, with the architecture assumptions we

made, such a network can indeed be constructed. In particular, the architectural assumptions allow us to make independent copies of the output of the stochastic block and the linear transformation after it allows us to average over such independent copies to recover the mean with a vanishing variance, which can then be shown to be able to achieve zero loss. □

**Remark.** *The condition that the width is a multiple of $d^*$ is not essential and is only used for making the proof concise. One can prove the same result without requiring $d_1 = kd^*$. Also, the dimension of $y$ is not essential here. When $y$ is high-dimensional, one can prove that for all $i$, $Var_\epsilon \left[ {}^{kd^*} f_{w_*,i}(x,\epsilon) \right]$ converges to zero.*

At a high level, one might wonder why the optimized model achieves zero variance. Our results suggest that the form of the loss function may be crucial. The MSE loss can be decomposed into a bias term and a variance term:

$$\sum_i^N L_i = bias + variance. \tag{6}$$

Minimizing the MSE loss involves both minimizing the bias term and the variance term, and the key step of our proof involves showing that a neural network with a sufficient width can reduce the variance to zero. We thus conjecture that convergence to a zero-variance model can be generally true for a broad class of loss functions. For example, one possible candidate for this function class is the set of convex loss functions, which favor a mean solution more than a solution with variance (by Jensen's inequality), and a neural network is then encouraged to converge to such solutions so as to minimize the variance. However, identifying this class of loss functions is beyond the scope of the present work, and we leave it as an important future step. Lastly, we also stress that the main results are not a trivial consequence of the MSE being convex. When the model is linear, it is rather straightforward to show that the variance reduces to zero in the large-width limit because taking the expectation of the model output is equivalent to taking the expectation of the latent noise. However, this is not trivial to prove for a neural network because the net $f$ is, in general, a nonlinear and nonconvex function of the latent noise $\epsilon$.

While our result does not directly deal with the predictive variance of the model on the testing set (or other out-of-distribution data), our results do have some implication on the behavior of the out-of-distribution data points when viewed together with the result of Foong et al. (2020). Foong et al. (2020) shows that, in many regions of the input space, the prediction variance of the model of any interpolation of two points is bounded by the sum of the variance on each of these points. This implies that our results may also be quite relevant for the points not in the training set, explaining our empirical observation (See Section 4) that the variance on the test points also drops with the variance on the training set.

### 3.3 Application to Dropout

**Definition 4.** A stochastic block $g(x)$ is said to be a *p*-dropout layer if $[g(x)]_j = \epsilon_j[h(x)]_j$, where $h(x)$ is a deterministic block, and $\epsilon_j$ are independent random variables such that $\epsilon_j = 1/p$ with probability $p$ and $\epsilon_j = 0$ with probability $1 - p$.

Since the noise of dropout is independent, one can immediately apply the main theorem and obtain the following corollary.

**Corollary 1.** *For any $0 < p < 1$, an optimized stochastic network with an infinite width p-dropout layer has zero variance on the training set.*

Our result thus formally proves the intuitive hypothesis in the original dropout paper (Srivastava et al., 2014) that applying dropout to training has the effect of encouraging an averaging effect in the latent space.

### 3.4 Convergence with a Stochasticity Regularization

In many scenarios, we want to train a stochastic network with regularization that is, say, due to a prior term in the loss function. We now extend our result to the case where a soft constraint exists in the loss function; such a constraint often appears in Bayesian deep learning to encourage the learnable latent variables to

conform to the desired distribution. The result in this section is more general and involved than Theorem 1 because the soft prior constraint matches the latent variable to the prior distribution in addition to the MSE loss. The existence of the prior term regularizes the model and prevents a perfect fitting of the original MSE loss, and the main message of this section is to show that even in the presence of such a (weak) regularization term, we can still have a vanishing variance, but its decay toward zero is relatively slow.

While it is natural that a latent variable model $f = g^2 \circ g^1$ can be decomposed into two blocks, where $g^2$ is the decoder, and $g^1$ is an encoder, we require one additional condition. Namely, the stochastic block ends with a linear transformation layer.[4]

**Definition 5.** A stochastic block $g(x)$ is said to be a encoder block if $g(x) = W_1 h(x) + \epsilon_j \odot (W_2 h(x) + b)$, where $\odot$ is the Hadamard product, $W_1$, $W_2$ are linear transformations and $b$ is the bias of $W_2$, $h(x)$ is a deterministic block, and $\epsilon_j$ are uncorrelated random variables with zero mean and unit variance.

Note that we explicitly require the weight matrix $W_2$ to come with a bias. For the other linear transformations, we have omitted the bias term for notational simplicity. One can check that this definition is consistent with Definition 2. Namely, a net with an encoder block is indeed a type of stochastic net. When the training loss only consists of the MSE, it should follow as an immediate corollary of Theorem 1 that the variance converges to zero. However, the prior term complicates the problem. The following definition specifies the type of the prior loss under our consideration.

**Definition 6.** (Prior-Regularized Loss.) Let $f = g^2 \circ g^1$, such that $[g^1(x)]_j = W_1 h(x) + \epsilon_j \odot W_2 h(x)$ is a encoder block. A loss function $\ell$ is said to be a prior-regularized loss function if $\ell = \sum_i L_i + \ell_{\text{prior}}$, where $\sum_i L_i$ is given by equation 3 and $\ell_{\text{prior}} = \frac{1}{d_1^\alpha} \sum_j \ell_{\text{mean}}([W_1 h(x_i)]_j) + \ell_{\text{var}}([W_2 h(x) + b]_j)$, where $\alpha > 0$, $\ell_{\text{mean}} \geq 0$ and $\ell_{\text{var}} \geq 0$ are differentiable functions that are equal to zero if for all $x_i$, $[W_1 h(x_i)]_j = 0$ and $[W_2 h(x) + b]_j = 1$.

We have abstracted away the actual details of the definitions of the prior loss. For our purpose, it is sufficient to say that the equation $[W_1 h(x_i)]_j = 0$ means that the loss function encourages the posterior to have a zero mean and $[W_2 h(x) + b]_j = 1$ encourages a unit variance. As an example, one can check that the standard ELBO loss for VAE satisfies this definition. With this architecture, we prove a similar result. The proof is given in Section B.2.

**Theorem 2.** *Assuming that the neural networks under consideration satisfy Assumptions 1 ,2, 4, and the stochastic block is a encoder block and satisfies Assumption 3, and that the loss function is a prior-regularized loss with parameter $\alpha > 0$, let $d_2, d_0$ be fixed integers, $^{d_1}f = {}^{d_2,d_1}g^2 \circ {}^{d_1,d_0}g^1$ and $\{{}^{d_1}f\}_{d_1 \in \mathbb{Z}^+}$ be a sequence of stochastic networks with stochastic block $^{d_1,d_0}g^1_{w(g_1)} \in \mathcal{S}(g^1)$. Let $^{d_1}f$ be overparameterized for all $d_1 \geq d^*$ for some $d^* > 0$. Let $w_* = \arg\min_w \sum_i^N \mathbb{E}[L(f_w^{d_i}(x, \epsilon), y_i)]$ be the global minimum of the loss function. Then, for all $x$ in the training set,*

$$\lim_{k \to \infty} Var_\epsilon \left[ {}^{kd^*} f_{w_*}(x, \epsilon) \right] = 0. \tag{7}$$

We stress that the scaling of the regularization strength as $d_1^{-\alpha}$, where $\alpha > 0$, is both relevant and important. Our result is actually complementary to the result of Coker et al. (2021), which shows that if $\alpha \leq 0$, the loss due to the regularization term indeed increases as $d_1$ as we increase the width towards infinity and, ultimately, the regularization becomes so strong that the model completely ignores the data and cannot learn anything at all. The only meaningful limit for the model is thus $\alpha > 0$. In fact, if the value of the regularization scales as $d_1$, the regularization strength needs to scale as $\alpha \geq 1$ for the regularization effect to not be infinitely strong. In light of Coker et al. (2021), our result thus suggests that the stochastic networks have an interesting bifurcative behavior at $\alpha = 0$: to the one side, the model cannot learn anything; to the other side, the model fits the data perfectly well.

One might also wonder whether a vanishing prior strength makes this setting trivial. The proof shows that it is far from trivial because we are simultaneously scaling up the model width and the regularization strength. In this case, even a prior with vanishing strength can have a very strong influence. The proof suggests that if the prior strength decays as $1/d_1^\alpha$, the variance should decay roughly as $d_1^{-\frac{\min(1,\alpha)}{2}}$. Namely, the smaller the $\alpha$, the slower the rate of convergence towards 0. Additionally, the fastest exponent at which the variance can

---

[4]Such a condition is satisfied by a standard VAE encoder (Kingma & Welling, 2013), for example.

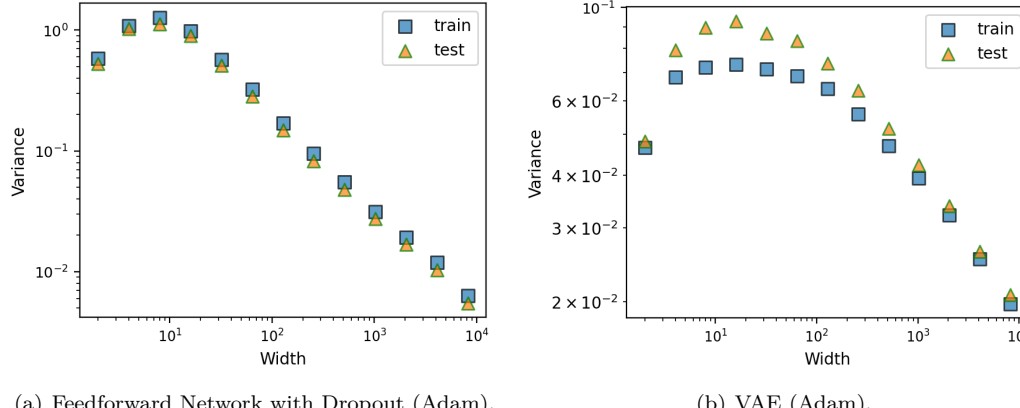

(a) Feedforward Network with Dropout (Adam).   (b) VAE (Adam).

Figure 2: Scaling of the prediction variance of different models as the width of the stochastic layer extends to infinity. For both dropout network and VAE, we see that the prediction variance decreases towards 0 as the width increases. For completeness, the prediction variance over an independently sampled test set is also shown.

decay is $-0.5$, significantly smaller than the case for Theorem 1, where the proof suggests an exponent of $-1$. This means that even a vanishing regularization strength can have a strong impact on the prediction variance of the model. Quantitatively, as $\alpha$ approaches zero, the variance can decay arbitrarily slowly. Qualitatively, our result implies that having a regularization term or not qualitatively changes the nature of the global minima of the stochastic model.

### 3.5 What causes a vanishing variance?

Now, we are ready to answer the major questions we raised in the introduction. First of all, *does a stochastic neural network have a vanishing variance when it is wide*? The answer is yes if the model has a "two-block" structure and we have successfully trained the model to one of its global minima. A two-block model can do a rather special operation: use the first block to make independent copies of the noisy latent variables and use the second block to average. We showed that while a trained model does not have to rely on this strategy for noise canceling, it must do worse as long as it is well-optimized.

The second question is more important: *what causes the noise to vanish*? In this case, our result shows that the common belief is not the full picture. Having a powerful decoder is not required for a vanishing variance. The key assumption in our proof is that the corresponding deterministic model is sufficiently "powerful" to memorize all the data points, and this can be achieved in one of the following three ways: (1) having a powerful decoder, which is the conventional understanding; (2) having a powerful encoder; (3) neither the encoder nor the decoder is powerful, but when taken together, they can memorize all the data points. Additionally, our work also gives a mathematical definition of "power" in the context of stochastic neural networks: a model is powerful if its deterministic version can memorize all the training data.

### 3.6 Practical Implication

Practically, our theory suggests that one may be able to prevent a vanishing variance in one of the following ways: (1) use a small and thin model; while this is often the preferred practice, it often leads to insufficient modeling of the data due to limited model capacity; (2) use data augmentation with large models; intuitively, this seems to be the better answer, with data augmentation, it often becomes impossible for the model to completely memorize the dataset, and yet, one can use a sufficiently large and expressive model to model the complicated nonlinearities in the data. We leave this as future work.

## 4 Numerical Simulations

We perform experiments with nonlinear neural networks to demonstrate the studied effect of vanishing variance. The first part describes the illustrative experiment presented at the beginning of the paper. The second part experimentally demonstrates that a dropout network and VAE have a vanishing prediction variance on the training set as the width tends to infinity. Additional experiments performed with weight decay and SGD are presented in the appendix.

### 4.1 Illustration

In this experiment, we let the target function be $y = \sin(x) + \eta$ for $x$ uniformly sampled from the domain $[-3, 3]$. The target is corrupted by a weak noise $\eta \sim \mathcal{N}(0, 0.1)$. The model is a feedforward network with tanh activation with the number of neurons $1 \to d \to 1$ where $d \in \{10, 50, 500\}$, and a dropout with dropout probability $p = 0.1$ is applied in the hidden layer. See Figure 1.

### 4.2 Dropout

We now systematically explore the prediction variance of a model trained with dropout. As an extension of the previous experiment, we let both the input and the target be vectors such that $x, y \in \mathbb{R}^d$. The target function is $y_i = x_i^3 + \eta_i$ for $i \in \{1, ..., d\}$, where $x_i, \eta_i \sim \mathcal{N}(\mathbf{0}, 1)$ and the noise $\eta \in \mathbb{R}^d$ is also a vector. We let $d = 20$ and sample 1000 data point pairs for training. The model is a three-layer MLP with ReLU activation functions, with the number of neurons $20 \to d_h \to 20$, where $d_h$ is the width of the hidden layer. Dropout is applied to the post-activation values of the hidden layer. In the experiments, we set the dropout probability $p$ to be 0.1, and we independently sample outputs 3000 times to estimate the prediction variance. Training proceeds with Adam for 4500 steps, with an initial learning rate of 0.01, and the learning rate is decreased by a factor of 10 every 1500 step. See Figure 2(a) for a log-log plot of width vs. prediction variance. We see that the prediction variance of the model on the training set decreases towards zero as the width increases, as our theory predicts. For completeness, we also plot the prediction variance for an independently sampled test set for completeness. We see that, for this task, the prediction variance of the test points agree well with that of the training set. A linear regression on the slope of the tail of the width-variance curve shows that the variances decrease roughly as $d^{-0.7}$, close to what our proof suggests ($d^{-1}$); we hypothesize that the exponent is slightly smaller than 1 because the training is stopped at a finite time and the model has not fully reached the global minimum.[5]

### 4.3 Variational Autoencoder

In this section, we conduct experiment on a $\beta$–VAE with latent Gaussian noise. The input data $x \in \mathbb{R}^d$ is sampled from a standard Gaussian distribution with $d = 20$. We generate 100 data points for training. The VAE employs a standard encoder-decoder architecture with ReLU nonlinearity. The encoder is a two-layer feedforward network with neurons $20 \to 32 \to 2 \times d_h$. The decoder is also a two-layer feedforward network with architecture $d_h \to 32 \to 20$. Note that our theory requires the prior term $\ell_{\text{vae}}$ not to increase with the width; we, therefore, choose $\beta = 0.1/d_h$. The training objective is the minus standard Evidence Lower Bound (ELBO) composed of reconstruction error and the KL divergence between the parameterized variable and standard Gaussian. We independently sample outputs 100 times to estimate the prediction variance for estimating the variance. The results in Fig. 2(b) show that the variances of both training and test set decrease as the width increases and follow the same pattern.

### 4.4 Experiments with Weight Decay

Weight decay often has the Bayesian interpretation of imposing a normal distribution prior over the variables, and sometimes it is believed to prevent the model from making a deterministic prediction (Gal & Ghahramani, 2016). We, therefore, also perform the same experiments as in the previous two subsections, with a weight decay strength of $\lambda = 5e - 4$. See Appendix Sec. A.1 for the results. We notice that the experimental result

---

[5]Moreover, with a finite-learning rate, SGD is a biased estimator of a minimum (Ziyin et al., 2021a).

is similar and that applying weight decay is not sufficient to prevent the model from reaching a vanishing variance. Conventionally, we expect the prediction of a dropout net to be of order $\lambda$ and not directly dependent on the width (Gal & Ghahramani, 2016); however, this experiment suggests that width may wield a stronger influence on the prediction variance than the weight decay strength. Since a model cannot reach the actual global minimum when weight decay is applied, this experiment is beyond the applicability of our theory; therefore, this result suggests a conjecture that even in a local minimum, it can still be highly likely for stochastic models to reach a solution with vanishing variance, and proving or disproving this conjecture can be an important future task.

### 4.5 Other loss functions

Empirically, we note that neural networks trained with other common loss functions, such as the cross-entropy loss, also exhibit the same phenomenon. We experiment on the MNIST dataset with two architectures: (1) a fully connected neural network with architecture $784 \rightarrow width \rightarrow 10$ neurons (labeled as *FC*) and (2) a simple convolutional neural network with two convolution layers and two fully connected tanh layers with architecture: *convolution* $\rightarrow$ *convolution* $\rightarrow$ *width* $\rightarrow$ 10. In both cases, we apply a dropout of probability 0.3 to the neurons with dimension *width* and train with Adam on the cross-entropy loss until the loss stops decreasing. See Figure 3. We see that the predictive variance also decreases toward zero, similar to the case of the MSE loss.

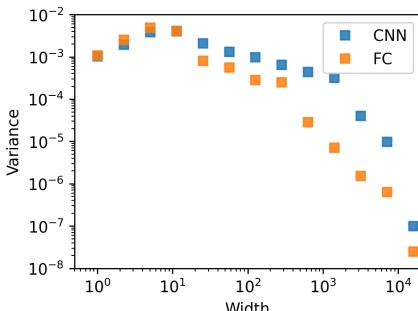

Figure 3: Predictive variances of two models trained on MNIST with the cross-entropy loss decrease towards zero as the width increases.

## 5 Discussion

In this work, we studied the potential causes for the prediction variance of stochastic neural networks to vanish. We showed that when the loss function satisfies certain conditions and under mild architectural conditions, the prediction variance of a stochastic net on the training set tends to zero as the width of the stochastic layer tends to infinity. Our theory offers a precise mathematical explanation to a frequently quoted anecdotal problem of a stochastic network, that the neural networks can be too powerful such that adding noise to the latent layers is not sufficient to make a network capable of modeling a distribution well (Higgins et al., 2016; Burgess et al., 2018; Dai & Wipf, 2019). A major limitation of our theory is that we have only studied the global minimum, and it is important to study whether or under what conditions the variance also vanishes for a local minimum. From a practical point of view, our result suggests that it is generally nontrivial to train a model whose prediction variance matches the true variance of the data. Our results thus motivate the design loss functions that encourage a nonvanishing prediction variance that matches the desired level of uncertainty, which is an interesting future problem.

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

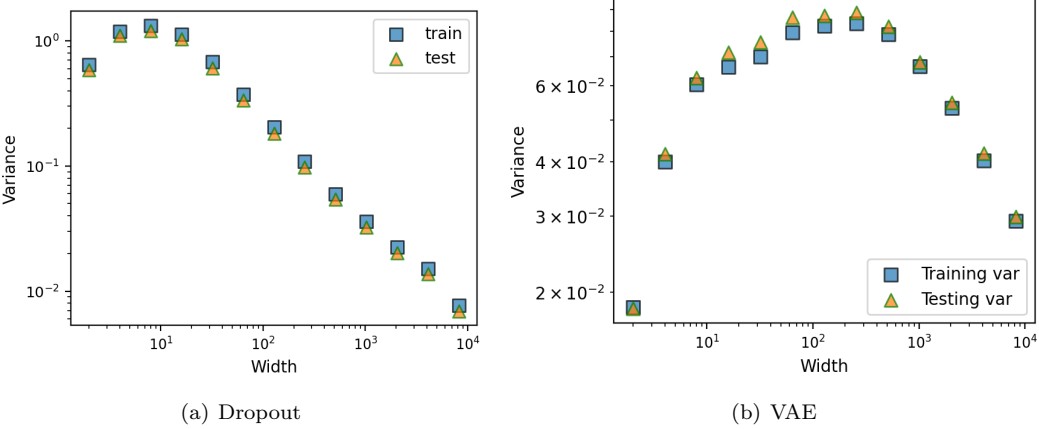

(a) Dropout

(b) VAE

Figure 4: Variance vs. the width of NNs using weight decay.

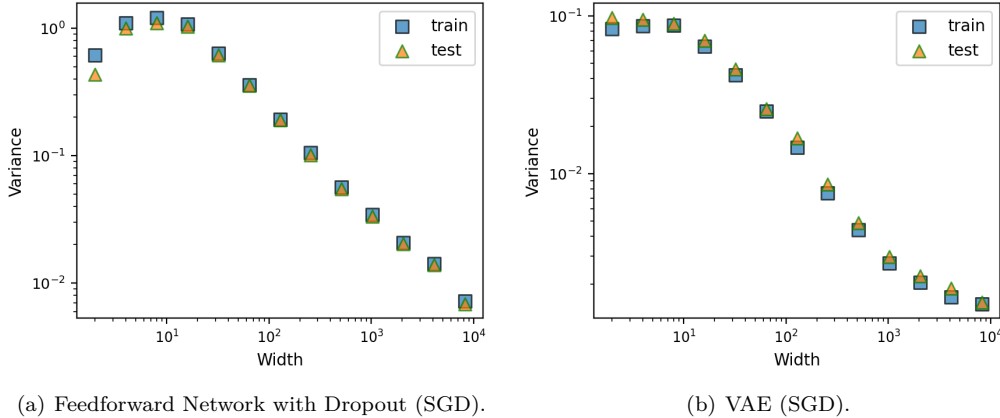

(a) Feedforward Network with Dropout (SGD).

(b) VAE (SGD).

Figure 5: Empirical scaling of the prediction variance of different models as the width of the stochastic layer extends to infinity. For both dropout network and VAE, we see that the prediction variance decreases towards 0 as the width increases.

## A Additional Experiments

### A.1 Weight decay

This part of experiment has been described in the main text. See Figure 4. We see that even with weight decay, the prediction variance drops towards zero unhindered.

### A.2 Training with SGD

Since our result only depends on the global minimum of the loss function, one expects to also find that the prediction variance to decrease with a different optimization procedure. In this section, we perform the same experiment with SGD. See Figure 5. We see that, for dropout, the result is similar to the case with Adam. For VAE, the result is a little more subtle in the tail, where the decrease in variance slows down. We hypothesize that it is because the SGD algorithm increase in fluctuation and reduce in stability as the width of the hidden layer increases (Liu et al., 2021; Ziyin et al., 2021b), which causes the prediction variance to increase and partially offsets the effect due to averaging of the parameters.

# B Proofs and Theoretical Concerns

## B.1 Proof of Theorem 1

*Proof.* In the proof, we denote the term $L\big({}^{d_1}f_w(x,\epsilon),y_i\big)$ as $L_i^{d_1}(w)$. Let $w_*$ be the global minimizer of $\sum_j^N \mathbb{E}_\epsilon\big[L_j^{d_1}(w)\big]$. Then, for any $w$, we have by definition

$$0 \le \sum_j^N \mathbb{E}_\epsilon\big[L_j^{d_1}(w_*)\big] \le \sum_j^N \mathbb{E}_\epsilon\big[L_j^{d_1}(w)\big]. \tag{8}$$

If $\lim_{d_1\to\infty}\sum_j^N \mathbb{E}_\epsilon\big[L_j^{d_1}(w)\big] = 0$, we have $\lim_{d_1\to\infty}\sum_j^N \mathbb{E}_\epsilon\big[L_j^{d_1}(w_*)\big] = 0$, which implies that $\lim_{d\to\infty}\mathbb{E}_\epsilon\big[L_j^{d_1}(w_*)\big] = 0$ for all $j$. By the bias-variance decomposition of the MSE, this, in turn, implies that $\mathrm{Var}\big[{}^{d_1}f_{w_*}(x_j)\big] = 0$ for all $j$. Therefore, it is sufficient to construct a sequence of $w$ such that $\lim_{d_1\to\infty}\sum_j^N \mathbb{E}_\epsilon\big[L_j^{d_1}(w)\big] = 0$.

Now, we construct such a $w$. Let ${}^{d_1}f'$ denote the deterministic counterpart of ${}^{d_1}f$. By definition and by Assumption 1, we have

$$^{d_1}f(x) = {}^{d_2,d_1}g_{w^2}^2 \circ {}^{d_1,d_0}g_{w^1}^1(x); \tag{9}$$

$$^{d_1}f'(x) = {}^{d_2,d_1}g_{w^2}^2 \circ \mathbb{E}_\epsilon \circ {}^{d_1,d_0}g_{w^1}^1(x). \tag{10}$$

By the architecture assumption (assumption 4), there exists a function $h_v$ parametered by a parameter set $v$ and a linear transformation $M$ such that we can further decompose the two neural networks as

$$f(x) = h_v \circ M \circ g_{w_1}^1(x); \tag{11}$$

$$f'(x) = h_v \circ M \circ \mathbb{E} \circ g_{w_1}^1(x), \tag{12}$$

where $M \in \mathbb{R}^{d\times d_1}$ for a fixed integer $d$ and is a linear transformation belonging to the parameter set of $g^2$. Note that, by definition, the parameter set of the $g^2$ block is $w_2 = v \cup M$.

Let $u_*$ be a global minimum of ${}^{d_1}f'$:

$$u_* := (v_*, M_*', w_*^1) = \arg\min_{v,M',w_1} \sum_j^N L\big({}^{d_1}f_w'(x),y_i\big), \tag{13}$$

and, by the assumption of overparametrization, we also have ${}^{d_1}f_{u_*}'(x_j) = y_j$ for all $j$.

We now specify the parameters for $f_w$ for $k > 1$. By assumption 2, we can find parameters $w_1'$ such that $\mathbb{E}\big[{}^{kd^*,d_0}g_{w_1'}^1(x)_j\big] = \mathbb{E}\big[{}^{d^*,d_0}g_{w_*}^1(x)\big]_{j \bmod d^*}$ for $j = 1,...,kd^*$. Namely, we choose the parameters such that the expected output of the stochastic block are $k$ identical copies of the output of the overparametrized deterministic model with width $d^*$.

Since $M$ is a linear transformation, one can factorize it as a product of two matrices such that $M = AG$, where $A \in \mathbb{R}^{d\times d^*}$ and $G \in \mathbb{R}^{d^*\times d_1}$:

$$f(x) = h_v \circ A \circ G \circ g^1(x)_{w_1}; \tag{14}$$

$$f'(x) = h_v \circ A \circ G \circ \mathbb{E} \circ g_{w_1}^1(x). \tag{15}$$

Now, note that by definition, the function $h_v \circ A$ for any ${}^{kd^*}f$ coincides with the $g_2$ block of ${}^{d^*}f'$. Namely, $h_v \circ A = {}^{d_2,d^*}g_{w^2}^2$ such that $w^2 = v \cup A$, and we let $v = v^*$ and $A = M^*$.

Now, the last step is to specify $G$. We let $G_{ij}^* = \frac{1}{k}\delta_{i,j \bmod d}$, where $\delta_{i,j} = 1$ if $i = j$ and 0 otherwise. Namely, $G^*$ is nothing but an averaging matrix that sums and rescales the deterministic layer by a factor of $1/k$.

To summarize, our specification defines the following stochastic neural network:

$$^{kd*}f(x) = h_{v^*} \circ M^* \circ G^* \circ g_{w_1'}^1(x). \tag{16}$$

By definition of $G^*$ and $w_1'$, $\mathbb{E}_\epsilon[G^* \circ {}^{kd^*,d_0}g^1_{w_1'}(x)] = \mathbb{E}_\epsilon[{}^{d^*,d_0}g^1_{w_1'}(x)]$, and $[G^* \circ {}^{kd^*,d_0}g^1_{w_1'}(x)]_j$ are independent for different $j$. Moreover, because $[{}^{d^*,d_0}g^1_{w_1'}(x)]_j$ has variance $\Sigma_{ii}$ by assumption, $[G^* \circ {}^{kd^*,d_0}g^1_{w_1'}(x)]_j$ has variance $\Sigma_{ii}/k$.

Now, as $k \to \infty$,

$$G^* \circ {}^{kd^*,d_0}g^1_{w_1'}(x) \to_{L^2} \mathbb{E}_\epsilon[G^* \circ {}^{kd^*,d_0}g^1_{w_1'}(x)], \tag{17}$$

where $\to_{L^2}$ denotes convergence in mean square. Because $h_v \circ A$ is a Lipshitz continuous function by Assumption 1,

$$\lim_{k\to\infty} h_{v^*} \circ M^* \circ G^* \circ {}^{kd^*,d_0}g^1_{w_1'}(x) \to_{L^2} h_{v^*} \circ M^* \circ \mathbb{E}_\epsilon \circ {}^{d^*,d_0}g^1_{w_1^*}(x). \tag{18}$$

This implies that the expectation of the constructed model with increasing width converge to that of the overparametrized determinstic model (convergence in mean square implies convergence in mean). Therefore, defining our model as ${}^{d_1}f^* = h_{v^*} \circ M^* \circ G^* \circ {}^{kd^*,d_0}g^1_{w_1'}$, we obtain

$$\mathbb{E}_\epsilon[{}^{d_1}f(x_i)] \to {}^{d^*}f'(x_i). \tag{19}$$

Therefore, we have, by the bias-variance decomposition for MSE:

$$\mathbb{E}_\epsilon[{}^{d_1}f^*(x_i,\epsilon) - y_i]^2 = \left[\mathbb{E}_\epsilon[{}^{d_1}f^*(x_i,\epsilon)] - y_i\right]^2 + \mathrm{Var}[{}^{d_1}f^*(x_i,\epsilon)] \tag{20}$$

Both terms converges to 0 for all $i$, and so the sum of two terms converge to 0. This finishes the proof. □

## B.2  Proof of Theorem 2

Before the proof, we first comment that the proof is quite similar to the previous case. The difference lies in how we construct the model so as to reduce the training loss to zero as

*Proof.* In the proof, we denote the term $L({}^{d_1}f_w(x,\epsilon), y_i)$ as $L_i^{d_1}(w)$. Let $w_*$ be the global minimizer of $\sum_j^N \mathbb{E}_\epsilon\left[L_j^{d_1}(w)\right]$. Then, for any $w$, we have by definition

$$0 \le \sum_j^N \mathbb{E}_\epsilon\left[L_j^{d_1}(w_*)\right] \le \sum_j^N \mathbb{E}_\epsilon\left[L_j^{d_1}(w)\right]. \tag{21}$$

If $\lim_{d_1\to\infty} \sum_j^N \mathbb{E}_\epsilon\left[L_j^{d_1}(w)\right] = 0$, we have $\lim_{d_1\to\infty} \sum_j^N \mathbb{E}_\epsilon\left[L_j^{d_1}(w_*)\right] = 0$, which implies that $\lim_{d\to\infty} \mathbb{E}_\epsilon\left[L_j^{d_1}(w_*)\right] = 0$ for all $j$. This, in turn, implies that $\mathrm{Var}[{}^{d_1}f_{w_*}(x_j)] = 0$ for all $j$ because both the reconstruction loss and the prior are non-negative. Therefore, it is sufficient to construct a sequence of $w$ such that $\lim_{d_1\to\infty} \mathbb{E}_\epsilon \sum_j^N \left[L_j^{d_1}(w)\right] = 0$.

Let ${}^{d_1}f'$ denote the deterministic counterpart of ${}^{d_i}f$, and let $u_*$ be a global minimum of ${}^{d_1}f'$. By the definition of a neural network, we can write

$$\begin{aligned}
{}^{d_1}f(x) &= {}^{d_2,d_1}g^2_{w^2} \circ {}^{d_1,d_0}g^{w^1}(x); \tag{22}\\
{}^{d_1}f'(x) &= {}^{d_2,d_1}g^2_{w^2} \circ \mathbb{E}_\epsilon \circ {}^{d_1,d_0}g^{w^1}(x). \tag{23}
\end{aligned}$$

By the assumption of overparametrization, ${}^{d_1}f'_{u_*}(x_j) = y_j$ for all $j$.

By the architecture assumption, there exists a function $h_v$ parametrized by a parameter set $v$ such that we can further decompose the two neural networks as

$$\begin{aligned}
f(x) &= h_v \circ M \circ Z \circ g_{w_1}(x); \tag{24}\\
f'(x) &= h_v \circ M \circ \mathbb{E} \circ Z \circ g_{w_1}(x), \tag{25}
\end{aligned}$$

where $M \in \mathbb{R}^{d_1 \times d}$ for a fixed integer $d$ and is a linear transformation belonging to the parameter set of $g^2$, and $Z(x) = T_1 x + \epsilon \odot (T_2 x + b)$ is the linear stochastic layer. Now, the parameter set of the network $f$ is $w = v \cup M \cup T_1 \cup T_2 \cup b \cup w_1$.

By definition, the loss takes the form

$$\sum_i^N L_i + \sum_j^{d_1} \ell_{\text{mean}}^{(j)} + \sum_j^{d_1} \ell_{\text{var}}^{(j)} \tag{26}$$

We first let $T_2 = 0$ and $b = 1$, which immediately minimizes the variance part of of the loss: $\ell_{\text{var}}^{(j)} = 0$ for all $j$. By assumption, for $d_1 \geq d^*$, $f'(x)$ is overparametrized, and one can find $(v_*, M_*', w_*^1, T_1^*) = \arg\min_{v,M',w_1} \sum_j^N L_j^{d^*}(w)$ such that $\sum_i^N L_i = 0$.

For $k > 1$, we let $[T_1']_{j:} = a[T_1^*]_j \mod d$ for a positive scalar $a$. Namely, we copy the rows of the matrix $W_1^*$ so that the expected output of the stochastic block are $k$ identical copies of the output of the overparametrized deterministic model with width $d^*$, rescaled by a factor $a$.[6]

Since $M$ is a linear transformation, one can factorize it as a product of two matrices such that $M = AG$ where $A \in \mathbb{R}^{d \times d^*}$ and $G \in \mathbb{R}^{d^* \times d_1}$:

$$f(x) = h_v \circ A \circ G \circ g^1(x)_{w_1}; \tag{27}$$

$$f'(x) = h_v \circ A \circ G \circ \mathbb{E} \circ g_{w_1}^1(x). \tag{28}$$

Now, by definition, the function $h_v \circ A = {}^{d_2,d^*}g_{w^2}^2$ such that $w^2 = v \cup A$, and we let $v = v^*$ and $A = M^*/a$. Again, notice that we have rescaled the matrix by a factor of $1/a$.

Now, the last step is specify $G$. We let $G_{ij}^* = \frac{1}{k}\delta_{i,j} \mod d$ where $\delta_{i,j} = 1$ if $i = j$ and 0 otherwise. Namely, $G^*$ sums and rescales the deterministic layer by a factor of $1/k$. This transformation has an averaging effect.

To summarize, our specification defines the following stochastic neural network:

$$f(x) = h_{v^*} \circ M^* \circ G^* \circ g_{w_1'}^1(x). \tag{29}$$

By definition of $G^*$ and $w_1'$, $\mathbb{E}_\epsilon[G^* \circ {}^{kd^*,d_0}g_{w_1'}^1(x)] = \mathbb{E}_\epsilon[{}^{d^*,d_0}g_{w_1'}^1(x)]$, and $[G^* \circ {}^{kd^*,d_0}g_{w_1'}^1(x)]_j$ are independent for different $j$. Moreover, because $[{}^{d^*,d_0}g_{w_1'}^1(x)]_j$ has variance $\Sigma_{ii}$ by assumption, $[G^* \circ {}^{kd^*,d_0}g_{w_1'}^1(x)]_j$ has variance $\Sigma_{ii}/(ak)$. We let $a = k^{-\gamma}$, where $1 > \gamma > 1 - \alpha$, and, therefore, $[G^* \circ {}^{kd^*,d_0}g_{w_1'}^1(x)]_j$ has variance $\Sigma_{ii}/(k^{1-\gamma})$ which vanishes to 0 as $k$ increases.

At the same time, because $\ell$ is a differentiable function of $T_1 x$,

$$\frac{1}{(kd^*)^\alpha} \sum_j^{kd^*} \ell_{\text{mean}}^{(j)} \sim \frac{1}{(kd^*)^\alpha} \sum_j^{kd^*} k^{-\gamma} + O(k^{-2\gamma}) \tag{30}$$

$$= k^{1-\gamma-\alpha}(d^*)^{1-\alpha} \tag{31}$$

$$\to 0, \tag{32}$$

where the last line follows from the condition $\gamma > 1 - \alpha$, which holds by assumption.[7]

Now, as $k \to \infty$,

$$G^* \circ {}^{kd^*,d_0}g_{w_1'}^1(x) \to_{L^2} \mathbb{E}_\epsilon[G^* \circ {}^{kd^*,d_0}g_{w_1'}^1(x)], \tag{33}$$

where $\to_{L^2}$ denotes convergence in mean square. Because $h_v \circ A$ is a Lipshitz continuous function,

$$\lim_{k\to\infty} h_{v^*} \circ M^* \circ G^* \circ {}^{kd^*,d_0}g_{w_1'}^1(x) \to_{L^2} h_{v^*} \circ M^* \circ \mathbb{E}_\epsilon \circ {}^{d^*,d_0}g_{w_1^*}^1(x). \tag{34}$$

This implies that the expectation of the model with increasing width converge to that of the overparametrized deterministic model (convergence in mean square implies convergence in mean. Therefore, defining our model as ${}^{d_1}f^* = h_{v^*} \circ M^* \circ G^* \circ {}^{kd^*,d_0}g_{w_1'}^1$

$$\mathbb{E}_\epsilon[{}^{d_1}f(x_i)] \to {}^{d^*}f'(x_i). \tag{35}$$

---

[6]Note that this factor of $a$ is one crucial difference from the previous proof. This factor of $a$ will be crucial for reducing $\ell_{\text{mean}}$ to 0.

[7]Namely, with this construction, the variance part of the loss scales as $k^{-(1-\gamma)}$ and the prior part of the loss scales as $k^{-(\gamma+alpha-1)}$. The sum of the two terms are minimized if $1 - \gamma = \gamma + alpha - 1$, or, $\gamma = 1 - \alpha/2$.

Therefore, defining our model as $^{d_1}f^* = h_{v^*} \circ M^* \circ G^* \circ {}^{kd^*,d_0}g^1_{w'_1}$, we have, by the bias-variance decomposition for MSE:

$$\mathbb{E}_\epsilon[^{d_1}f^*(x_i,\epsilon) - y_i]^2 = \left[\mathbb{E}_\epsilon[^{d_1}f^*(x_i,\epsilon)] - y_i\right]^2 + \text{Var}[^{d_1}f^*(x_i,\epsilon)], \tag{36}$$

which converges to 0. This finishes the proof. $\square$

## B.3  Removing the Bottleneck Constraint for $g^2$

In this section, we prove a version of Theorem 1 to demonstrate how one can remove the Bottleneck constraint of $g^2$. A similarly generalized version of Theorem 2 can also be proved using following the same steps, and so we leave that as an exercise to the readers. To begin, we first need an extended version of Assumption 5.

**Assumption 5.** (A model with larger width can express a model with smaller width II.) Additionally, let $x'$ denote a subset of $x$ (namely, $\dim(x') \leq \dim(x)$ and for all $i \in [1, \dim(x')]$, there exists $j$ such that $x_j = x'_i$). Let $g$ be a block and $\mathcal{S}(g)$ its block set. Each block $g = g_w$ in $\mathcal{S}(g)$ is associated with a set of parameters $w$ such that for any pair of functions $^{d_1,\dim(x)}g, {}^{d_1,\dim(x')}g' \in \mathcal{S}(g)$, any fixed $w'$, and any mappings $m$ from $\{1,...,d'_1\} \to \{1,...,d_1\}$, there exists parameters $w$ such that $^{d_1,\dim(x)}g_w(x) = {}^{d_1,\dim(x')}g_{w'}(x')$ for all $x'$.

The original Assumption 2 only specifies what it means to have a larger *output* dimension. This extended version, in addition, says what it means to have a larger input dimension for a block. We note that this additional condition is quite general and is satisfied by the usual structures, such as a fully connected layer. As the original Assumption 2, this assumption also agrees with the standard intuitive understanding of what it means to have a larger width.

With this additional assumption, we can remove the bottleneck requirement in the original Assumption 4. Formally, we now require the following weak condition for the architecture.

**Assumption 6.** ($g^2$ can be further decomposed into two blocks) Let $f = g^2 \circ g^1$ be the stochastic neural network under consideration, and let $g^1$ be the stochastic layer. We assume that for all $^{i,j}g \in \mathcal{S}(g^2)$, $^{i,j}g_w = g'_{w'}(Wx)$ for a block $g'$ with its block set $\mathcal{S}(g')$. $W$ is a linear transformation with the standard block set (see Proposition 1).

This generalized assumption effectively means that the model can be decomposed into three blocks:

$$f(x) = {}^{d_2,D}g' \circ {}^{D,d_1}W \circ {}^{d_1,d_0}g^1(x). \tag{37}$$

With these extended assumptions, we can prove a more general version of Theorem 1. In comparison to the original Theorem 1, this theorem effectively allows one to increase the width of all the layers that $g^1$ and $g^2$ implicitly contain simultaneously.

**Theorem 3.** *Let the neural network under consideration satisfy Assumptions 1 ,2, 5, 6 and 3, and assume that the loss function is given by equation 3. Let $\{^{d_1}f\}_{d_1 \in \mathbb{Z}^+}$ be a sequence of stochastic networks, for fixed integers $d_2, d_0$, $^{d_1}f = {}^{d_2,d_1}g^2 \circ {}^{d_1,d_0}g^1$ with stochastic block $^{d_1,d_0}g^1_{w(g_1)} \in \mathcal{S}(g^1)$. Additionally, let $D = D(d_1)$ be an monotonically increasing function of $d_1$ such that for $^{d_1}f = {}^{d_2,d_1}g^2 \circ {}^{d_1,d_0}g^1$,*

$$^{d_2,d_1}g^2 = {}^{d_2,D(d_1)}g' \circ {}^{D(d_1),d_1}W. \tag{38}$$

*Let $^{d_1}f$ be overparameterized for all $d_1 \geq d^*$ for some $d^* > 0$. Let $w_* = \arg\min_w \sum_i^N \mathbb{E}[L(^{d_i}f_w(x,\epsilon), y_i)]$ be a global minimum of the loss function. Then, for all $x$ in the training set,*

$$\lim_{k \to \infty} Var_\epsilon\left[^{kd^*}f_{w_*}(x,\epsilon)\right] = 0. \tag{39}$$

*Proof.* In the proof, we denote the term $L(^{d_1}f_w(x,\epsilon), y_i)$ as $L_i^{d_1}(w)$. Let $w_*$ be the global minimizer of $\sum_j^N \mathbb{E}_\epsilon\left[L_j^{d_1}(w)\right]$. Then, for any $w$, we have by definition

$$0 \leq \sum_j^N \mathbb{E}_\epsilon\left[L_j^{d_1}(w_*)\right] \leq \sum_j^N \mathbb{E}_\epsilon\left[L_j^{d_1}(w)\right]. \tag{40}$$

If $\lim_{d_1 \to \infty} \sum_j^N \mathbb{E}_\epsilon \left[ L_j^{d_1}(w) \right] = 0$, we have $\lim_{d_1 \to \infty} \sum_j^N \mathbb{E}_\epsilon \left[ L_j^{d_1}(w_*) \right] = 0$, which implies that $\lim_{d \to \infty} \mathbb{E}_\epsilon \left[ L_j^{d_1}(w_*) \right] = 0$ for all $j$. By the bias-variance decomposition of the MSE, this, in turn, implies that $\mathrm{Var}[^{d_1} f_{w_*}(x_j)] = 0$ for all $j$. Therefore, it is sufficient to construct a sequence of $w$ such that $\lim_{d_1 \to \infty} \sum_j^N \mathbb{E}_\epsilon \left[ L_j^{d_1}(w) \right] = 0$.

Now, we construct such a $w$. Let $^{d_1} f'$ denote the deterministic counterpart of $^{d_1} f$. By definition and by Assumption 1, we have

$$^{d_1} f(x) = {}^{d_2, d_1} g_{w^2}^2 \circ {}^{d_1, d_0} g_{w^1}^1(x); \tag{41}$$

$$^{d_1} f'(x) = {}^{d_2, d_1} g_{w^2}^2 \circ \mathbb{E}_\epsilon \circ {}^{d_1, d_0} g_{w^1}^1(x). \tag{42}$$

By the architecture assumption (assumption 6), there exists a function $h_v$ parametered by a parameter set $v$ and a linear transformation $M$ such that we can further decompose the two neural networks as

$$f(x) = h_v \circ M \circ g_{w_1}^1(x); \tag{43}$$

$$f'(x) = h_v \circ M \circ \mathbb{E} \circ g_{w_1}^1(x), \tag{44}$$

where $M \in \mathbb{R}^{d \times d_1}$ for a fixed integer $d$ and is a linear transformation belonging to the parameter set of $g^2$. Note that, by definition, the parameter set of the $g^2$ block is $w_2 = v \cup M$.

Let $u_*$ be a global minimum of $^{d_1} f'$:

$$u_* := (v_*, M_*', w_*^1) = \arg \min_{v, M', w_1} \sum_j^N L(^{d_1} f_w'(x), y_i), \tag{45}$$

and, by the assumption of overparametrization, we also have $^{d_1} f_{u_*}'(x_j) = y_j$ for all $j$.

We now specify the parameters for $f_w$ for $k > 1$. By assumption 2, we can find parameters $w_1'$ such that $\mathbb{E}[^{kd^*, d_0} g_{w_1'}^1(x)_j] = \mathbb{E}[^{d^*, d_0} g_{w_*^1}^1(x)]_{j \mod d^*}$ for $j = 1, ..., kd^*$. Namely, we choose the parameters such that the expected output of the stochastic block are $k$ identical copies of the output of the overparametrized deterministic model with width $d^*$.

Since $M$ is a linear transformation, we factorize it as a product of two matrices such that $M = AG$, where $A \in \mathbb{R}^{D(d_1) \times d^*}$ and $G \in \mathbb{R}^{d^* \times d_1}$:

$$f(x) = h_v \circ A \circ G \circ g^1(x)_{w_1}; \tag{46}$$

$$f'(x) = h_v \circ A \circ G \circ \mathbb{E} \circ g_{w_1}^1(x). \tag{47}$$

Now, note that by assumption 5, for any subset of any $x$, there exists $v'$ such that $h_{v'}(x) = h_{v*}'(x')$, where $h'$ is the corresponding block of $^{d^*} f'$. For $A$, we let the beginning columns of $A$ the same as $M^*$, and the remaining columns 0:

$$A = (M^* \quad 0). \tag{48}$$

With this choice, it follows that for any $x \in \mathbb{R}^{d^*}$, $h_{v'} \circ A(x) = g^2(x)$ for the $g^2$ block of $^{d^*} f'$.

Now, the last step is to specify $G$. We let $G_{ij}^* = \frac{1}{k} \delta_{i,j \mod d}$, where $\delta_{i,j} = 1$ if $i = j$ and 0 otherwise. Namely, $G^*$ is nothing but an averaging matrix that sums and rescales the deterministic layer by a factor of $1/k$.

To summarize, our specification defines the following stochastic neural network:

$$^{kd*} f(x) = h_{v*} \circ M^* \circ G^* \circ g_{w_1'}^1(x). \tag{49}$$

By definition of $G^*$ and $w_1'$, $\mathbb{E}_\epsilon[G^* \circ {}^{kd^*, d_0} g_{w_1'}^1(x)] = \mathbb{E}_\epsilon[^{d^*, d_0} g_{w_1'}^1(x)]$, and $[G^* \circ {}^{kd^*, d_0} g_{w_1'}^1(x)]_j$ are independent for different $j$. Moreover, because $[^{d^*, d_0} g_{w_1'}^1(x)]_j$ has variance $\Sigma_{ii}$ by assumption, $[G^* \circ {}^{kd^*, d_0} g_{w_1'}^1(x)]_j$ has variance $\Sigma_{ii}/k$.

Now, as $k \to \infty$,

$$G^* \circ {}^{kd^*,d_0}g^1_{w'_1}(x) \to_{L^2} \mathbb{E}_\epsilon[G^* \circ {}^{kd^*,d_0}g^1_{w'_1}(x)], \tag{50}$$

where $\to_{L^2}$ denotes convergence in mean square. Because $h_v \circ A$ is a Lipshitz continuous function by Assumption 1,

$$\lim_{k \to \infty} h_{v^*} \circ M^* \circ G^* \circ {}^{kd^*,d_0}g^1_{w'_1}(x) \to_{L^2} h_{v^*} \circ M^* \circ \mathbb{E}_\epsilon \circ {}^{d^*,d_0}g^1_{w^*_1}(x). \tag{51}$$

This implies that the expectation of the constructed model with increasing width converge to that of the overparametrized determinstic model (convergence in mean square implies convergence in mean). Therefore, defining our model as ${}^{d_1}f^* = h_{v^*} \circ M^* \circ G^* \circ {}^{kd^*,d_0}g^1_{w'_1}$, we obtain

$$\mathbb{E}_\epsilon[{}^{d_1}f(x_i)] \to {}^{d^*}f'(x_i). \tag{52}$$

Therefore, we have, by the bias-variance decomposition for MSE:

$$\mathbb{E}_\epsilon[{}^{d_1}f^*(x_i, \epsilon) - y_i]^2 = \left[\mathbb{E}_\epsilon[{}^{d_1}f^*(x_i, \epsilon)] - y_i\right]^2 + \text{Var}[{}^{d_1}f^*(x_i, \epsilon)] \tag{53}$$

Both terms converges to 0 for all $i$, and so the sum of two terms converge to 0. This finishes the proof. $\square$

