# OpenReview forum: "Why do wide stochastic neural networks have vanishing variance?"
_TMLR — Rejected by TMLR_

### Review · Reviewer_6ihi · 2023-01-08

**Summary Of Contributions:**

The paper studies stochastic neural networks, which is a family of functions $f(x,\varepsilon)$ that besides the regular data input $x$ takes additional random variable $\varepsilon$ as input. This family describes feed-forward networks with Dropout, or Variational Auto-Encoders (VAEs). The main object of study of the paper is the variance of the output distribution; namely, the variance of the random variable $f(x,\varepsilon)$, where we fix the data-point $x$ and transform the random variable $\varepsilon$ via $f(x,\cdot)$.

The main result of the paper is the following.
Assume that the stochastic network $f_{w^*}(x,\varepsilon)$ (with parameters $w$ and the random variable $\varepsilon$) attains the global optimum of the squared error over the dataset $\{(x_i, y_i)\}_{i=1}^N$, i.e.,

$w^* = \underset{w}{\operatorname{argmin}} \sum_i\mathbb{E}_{\varepsilon}\bigg(f_w(x_i,\varepsilon) - y_i\bigg)^2 \geq 0.$

The authors demonstrate that there exists such configuration of weights $w^*$ that the model attains zero loss, i.e.,

$\sum_i \mathbb{E}\varepsilon \bigg( f_{w^*} (x_i,\varepsilon) -  y_i \bigg)^2 = 0.$

This implies that the variance of the prediction is zero for every object in the dataset, i.e.,

$\mathbb{\text{Var}}_{\varepsilon}f(x_i,\varepsilon) = 0, \forall i\,.$

Namely, they construct such weights when the "width" (the dimensionality of the intermediate representation) goes to infinity.
The proposed construction is the following.
The initial linear layer outputs multiple copies of the same vector.
Then the stochastic part adds different noise to every copy.
The following linear layer averages all the copies out removing the noise and making the output completely deterministic.
Assuming that this procedure is expressive enough, the network then is able to achieve zero loss, which immediately implies zero variance of the outputs.

The authors claim that this result explains the vanishing variance of the predictions with the increasing width of a stochastic neural network. They also demonstrate the variance vanishing empirically on synthetic data. Namely, they train MLPs on the data from 20-dimensional standard normal gaussian (1000 points for the dropout model, 100 points for VAE).

**Audience:**

No

**Broader Impact Concerns:**

No discussion of the broader impact is needed.

**Claims And Evidence:**

No

**Requested Changes:**

I would suggest the authors to adjust their conclusions according to their results. I don't think that the paper reveals the actual cause of the vanishing variance, as stated here

```
The main implication of our result is that the popular belief that the strong expressivity of large neural networks causes the neural network causes prediction variance to vanish is a misattribution. We show that the actual cause of the vanishing effect is because deep neural
networks actually have a “two-layer" structure, where the upper layer can copy independent
copies of the latent variable, and the second layer can average over such copies to cancel
the noise.
```

The paper rather demonstrates that such weights exist when the network is "wide", but neither proves that it happens because of the growing width nor demonstrates that wide neural networks converge to the proposed construction.

Other examples of claims that should be adjusted
```
We show that the actual cause of the vanishing effect is because deep neural networks actually have a “two-layer" structure, where the upper layer can copy independent copies of the latent variable, and the second layer can average over such copies to cancel the noise.
```

```
First of all, does a stochastic neural network have a vanishing variance when it is wide? The answer is yes if the model has a “two-block" structure.
```

```
what causes the noise to vanish? In this case, our result shows that the common belief is not correct. A model does not need to have a strong expressivity to have a vanishing variance at all. All the model needs are the minimal capability of making redundant copies and averaging over such noises.
```

Also, I think the discussion of this claim is necessary
```
the popular belief that the strong expressivity of large neural networks causes the neural network causes prediction variance to vanish
```
since the authors also assume some notion of "expressivity" in their theorems.

**Strengths And Weaknesses:**

I think that the claims made in the paper are not aligned with the results. See the requests section for a detailed discussion.

The list of minor issues.
1. "In the related fields, A mainstream belief about a stochastic neural network is...". There is a typo, and this sentence needs to be supported with corresponding citations.
2. The following statements require citations and the definition of expressivity.
	- "widespread belief that the vanishing of variance of a wide neural network is caused by its expressivity"
	- "expressivity has nothing to do with a vanishing variance problem because a vanishing variance can happen even for linear models whose expressivity does not change"
3. Related works section does not relate the present paper to the literature. Instead, it mostly enumerates and briefly describes the literature from the field.
4. The definition of a block is not clear.
5. The definition of $\mathcal{S}(g)$ is not clear. Do the running indices $i,j$ correspond to the blocks with different input-output dimensions? Then how are they derived from the original block $g$?
6. Many sentences use unnecessary subjective modifiers: "good Bayesian inference can only be made if", "first actual layer", "we abuse the notation a little", "It is convenient to write", "for a genuine neural network".
7. Some sentences use unintroduced notions: "We now extend our result to the case where a soft constraint exists in the loss function", "a vanishing variance can still emerge, but relatively slowly".
8. Typo in Definition 6 in the equation referencing.
9. It's not clear what the authors mean by variance if the network outputs more than one dimension.

---

> ### Author Response · Authors · 2023-01-09
> **Quick clarification**
>
> Thanks for raising the clarifying question. We will make an update to the minor points you raised soon, but for now, let us clarify the most important point.
>
> First of all, we would like to point out that there is no mistake, and the proof is not trivial. Your confusion is due to a misunderstanding of definition 3 and our problem setting.
>
> First of all, the first condition you cited
> $$\lim_{d\to\infty} \sum_{i} \mathbb{E}_\epsilon(f(x_i,\epsilon)- y_i)^2 \quad (Eq. (A))$$
> is not an assumption at all but the goal of the proof.
>
> Secondly, you have misquoted definition 3. The actual definition 3 is this: let the model be written as $f=g^2  \circ g^1$, where $g^1$ is the stochastic block. Then, Def. 3 requires
> $$\sum_i( g^2 \circ \mathbb{E}_{\epsilon} \circ g^1 (x_i)- y_i)^2 =0 \quad (Eq. (B)).$$
>
> What the proof achieved is this: **assuming Eq. (B), prove Eq. (A)**.
>
> Note that this is not trivial at all. Because the second block $g^2$ is nonlinear in general, we have $ \mathbb{E}_\epsilon f(x\_i,\epsilon) \neq g^2 \circ \mathbb{E}\_{\epsilon} \circ g^1 (x_i);$
> therefore, without any further assumption on the function $f$,
> $$Eq. (B) \neq \sum_i \mathbb{E}_\epsilon (f(x_i,\epsilon) - y_i)^2,$$
> and so our result is not trivial.

---

> > ### Comment · Reviewer_6ihi · 2023-01-31
> > **Thank you for the clarification**
> >
> > Thank you for the clarification! I've updated the review.

---

### Review · Reviewer_VEVW · 2023-01-29

**Summary Of Contributions:**

This paper studies the vanishing variance phenomenon for stochastic neural networks. The main theorem states that a stochastic neural network having a "two-block" structure, under mild regularity conditions, will have zero variance as the width of the network grows to infinity. The high-level intuition behind this is that, in a two-block structure stochastic neural net, the first block can be used to construct independent copies of the noisy latents and the second block averages them out to achieve zero training error, and this operation basically reduces variances to zero. The theory is empirically validated for neural networks with dropouts and variational autoencoders.

**Audience:**

Yes

**Broader Impact Concerns:**

I don't have any concern on the ethical implications of the work.

**Claims And Evidence:**

Yes

**Requested Changes:**

- The theory is only for MSE loss, but in practice, we are often interested in other types of losses. One apparent example is cross-entropy loss. The authors already implied the possibility of extending the theory for a broader class of loss functions, so it would interesting to see at least an empirical result with the cross-entropy loss.
- I think the paper can be strengthened by adding some practical messages. As I stated earlier, it would be good to learn something from the main result, for instance, how should we design a stochastic neural net in order to avoid the pathological behavior explained in this paper. If possible, add the relevant discussion.


**Strengths And Weaknesses:**

Strength
- Presents a non-trivial theoretical analysis that clarifies popular misbeliefs that the vanishing variance problem of a stochastic neural network is due to its expressivity.
- The theory matches well with the empirical results.
- The paper is well-written.

Weaknesses
- As the authors already admitted, the result is only for the global minimum, while most of the modern neural networks are not guaranteed to achieve that. Also, the result says nothing about test variance, although it was empirically verified to hold for test data as well.
- Limited practical implications; so what should we do? If a network being too expressive is not the main cause of the vanishing variance problem, how should a stochastic neural network be designed?
- Only asymptotic results are given. For instance, in Figure (2), the widths of the networks where the variance vanishes are quite large, and in my opinion, these values are larger than the typical widths we use for stochastic neural networks. It would have been good to see how fast the variance decays as the width grows, not just the asymptotic result.

---

> ### Author Response · Authors · 2023-02-06
> **Author response**
>
> We thank the reviewer for the detailed and constructive assessment. In response to the review, we have included (1) additional experiments for the cross-entropy loss and (2) a discussion of the practical implications of our work. The major updates are colored in orange for emphasis.
>
>
> Specific answers to the requested changes and a weakness point are provided below.
>
>
> **"The theory is only for MSE loss, but in practice, we are often interested in other types of losses. One apparent example is cross-entropy loss. The authors already implied the possibility of extending the theory for a broader class of loss functions, so it would interesting to see at least an empirical result with the cross-entropy loss.
> "**
> - We now include experiments for a fully connected network and a convolutional neural network on MNIST trained with the cross-entropy loss in Section 4.5 of the updated version. The result shows that the predictive variance of these models also decreases to zero as we increase the width of the stochastic layer.
>
> **"I think the paper can be strengthened by adding some practical messages. As I stated earlier, it would be good to learn something from the main result, for instance, how should we design a stochastic neural net in order to avoid the pathological behavior explained in this paper. If possible, add the relevant discussion."**
>
> - Thanks for asking about this. We have included additional practical suggestions in Section 3.6. For example, we added the following discussion: "Practically, our theory suggests that one may be able to prevent a vanishing variance in one of the following ways: (1) use a small and thin model; while this is often the preferred practice, it often leads to insufficient modeling of the data due to limited model capacity; (2) use data augmentation with large models; intuitively, this seems to be the better answer, with data augmentation, it often becomes impossible for the model to memorize the dataset completely, and yet, one can use a sufficiently large and expressive model to model the complicated nonlinearities in the data."
>
> **"Only asymptotic results are given. For instance, in Figure (2), the widths of the networks where the variance vanishes are quite large, and in my opinion, these values are larger than the typical widths we use for stochastic neural networks. It would have been good to see how fast the variance decays as the width grows, not just the asymptotic result."**
> - We would like to humbly point out that we included both theoretical remarks and empirical discussions of the rate of decay of the variance. On the theory side, see the two paragraphs below Theorem 2. On the empirical side, see the discussion in Section 4.2.

---

### Review · Reviewer_g1xB · 2023-03-06

**Summary Of Contributions:**

The paper investigates why wide stochastic neural networks tend to have small variance in their prediction. This investigation is carried out for the mean squared loss for over-parameterized neural networks, specifically defined as those that reach the global minimum of the objective. In this setting, the Bias-variance decomposition shows that the variance term is necessarily zero. The assumptions and theorems in the paper mainly deal with constructing a stochastic network that is able to achieve the global minimum when the network width tends to infinity. Experiments are carried out for networks with dropout and variational autoencoders.

**Audience:**

Yes

**Claims And Evidence:**

Yes

**Requested Changes:**

See above

**Strengths And Weaknesses:**


Here are some of my concerns:

1. The following statement is made in the opening paragraph:
 "Since neural networks with stochastic latent layers are more difficult to train, stochasticity is believed to help regularize the model and prevent memorization of samples". This reasoning is inaccurate because training can be made difficult my making the network very deep. But such difficulties do not necessarily result in regularization, rather they are just optimization difficulties.
2. Regarding assumption 3: It is not clear that the covariance is taken over epsilon only, or both x and epsilon. Based on the sentence right below the assumption though, it seems like the expectation is only over epsilon. Also, the claim that this assumption is standard in stochastic techniques like the reparameterization trick, is a bit confusing. Specifically, in the reparameterization trick, we impose such a condition through the KL term, because the prior is assumed to be diagonal. I.e., this condition is not automatically satisfied by the network. So I pressume this assumption is made about a trained neural network, and not for a randomly initialized one. Is that the case?
3. There are two major limitations of the theorems in the paper:
- It is applicable when the loss value is precisely 0, i.e., it is not asymptotic in nature.
- The width needs to be infinite, i.e., there is no lower bound on the width beyond which the claim of small variance (more strictly 0 variance) holds true.
Both these limitations prevent the theorem from providing any useful insight in practical scenarios.
4. In section 3.6, it is implicitly assumed here that avoiding vanishing variance can be advantageous. What is the reasoning behind this?

In summary, the main idea of the paper is intuitive-- Bias-variance decomposition of a MSE loss function that reaches 0, shows that the variance term is necessarily zero. The authors use this to explain why over-parameterized models exhibit smaller variance in their output over the stochasticity. However, given that the expectation over the stochasticity (epsilon) is taken outside the MSE loss in the objective in Eq. 3, and it is assumed that this objective reaches the global minimum, i.e., 0, it is tautologically true that the prediction variance must be 0. So to sum it up, the assumption that the model is able to reach the global minimum itself is sufficient to imply that the model's prediction variance must be 0. This makes the theorems in the paper less interesting as they are merely focused on constructing a neural network that can achieve zero MSE loss as the network width tends to infinity (which in itself is not a surprising result).

---

> ### Author Response · Authors · 2023-03-18
> **Author response part 1**
>
> We thank the reviewer for the detailed feedback and criticisms. We think your main criticism is due to a misunderstanding of the overparametrization assumption. Please let us explain below.
>
> **"The following statement is made in the opening paragraph: "Since neural networks with stochastic latent layers are more difficult to train, stochasticity is believed to help regularize the model and prevent memorization of samples". This reasoning is inaccurate because training can be made difficult my making the network very deep. But such difficulties do not necessarily result in regularization, rather they are just optimization difficulties."**
> - We apologize for the inaccuracy. What you have suggested is what we intended. We have stressed that this is not necessarily true in the revision: "Since neural networks with stochastic latent layers are more difficult to train, stochasticity is *sometimes* believed to help regularize the model and prevent memorization of samples."
>
> **"Regarding assumption 3: It is not clear that the covariance is taken over epsilon only, or both x and epsilon. Based on the sentence right below the assumption though, it seems like the expectation is only over epsilon. Also, the claim that this assumption is standard in stochastic techniques like the reparameterization trick, is a bit confusing. Specifically, in the reparameterization trick, we impose such a condition through the KL term, because the prior is assumed to be diagonal. I.e., this condition is not automatically satisfied by the network. So I pressume this assumption is made about a trained neural network, and not for a randomly initialized one. Is that the case?"**
> - Thanks for pointing this out. The expectation is indeed taken over $\epsilon$ only, and so $\Sigma=\Sigma(x)$ remains a function of $x$. We have updated the assumption to make this clear.
>
> - We clarify that this assumption applies to the reparametrization trick in VAE in general, not just the trained models. In VAE, the hidden variable $h$ is generated as $h = \mu(x) + \sigma(x)\odot \epsilon$, where $\mu$ encodes the mean of $h$ and $\sigma(x)$ encodes the variance. Crucially, we often take the Hadamard product of $\sigma(x)$ and $\epsilon$. Immediately, one obtains that the covariance if $h$ is ${\rm diag}(\sigma^2(x))$, which is a diagonal matrix.
>
> **"It is applicable when the loss value is precisely 0, i.e., it is not asymptotic in nature."**
> - Thanks for pointing out this limitation. While it is true that we require the loss value for the deterministic model to be zero, we note that achieving a zero-training loss when there is no stochasticity is not too difficult. For example, performing simple linear regression on $n$ nondegenerate data points, each with dimension $d$, can achieve zero training loss as long as $d\geq n$. This is what it means to be "over-parametrized."
>
> **"The width needs to be infinite, i.e., there is no lower bound on the width beyond which the claim of small variance (more strictly 0 variance) holds true. Both these limitations prevent the theorem from providing any useful insight in practical scenarios."**
> - While the theorem statements only deal with the infinite width case, the actual proofs provide precise quantitative characterizations of how fast the variance decays as one increases the width. We humbly point out that these discussions have been presented in the manuscript and discussed in detail. On the theory side, see the two paragraphs below Theorem 2. On the empirical side, see the discussion in Section 4.2.
> - Also, knowing the rate of decay can be very practically useful. For example, one can run an experiment to determine what is the variance at a given width; one can then use the theory to estimate how much the variance will drop if one doubles the width.
>
> **"In section 3.6, it is implicitly assumed here that avoiding vanishing variance can be advantageous. What is the reasoning behind this?"**
> - We do not assume that avoiding a vanishing variance is advantageous. We only intended to say that if a practitioner wants to avoid a vanishing variance, one can try the following techniques we suggested. Whether avoiding a vanishing variance or not should be judged by the practitioner. In many cases, researchers do want to avoid a vanishing variance. For example, see https://arxiv.org/abs/1711.00464

---

> ### Author Response · Authors · 2023-03-18
> **Author response part 2**
>
> The following is the main criticism.
>
> **“However, given that the expectation over the stochasticity (epsilon) is taken outside the MSE loss in the objective in Eq. 3, and it is assumed that this objective reaches the global minimum, i.e., 0, it is tautologically true that the prediction variance must be 0. So to sum it up, the assumption that the model is able to reach the global minimum itself is sufficient to imply that the model's prediction variance must be 0. This makes the theorems in the paper less interesting as they are merely focused on constructing a neural network that can achieve zero MSE loss as the network width tends to infinity (which in itself is not a surprising result).”**
> - We clarify that we do not assume that the training loss in Eq. (3) can be reduced to zero. The goal of our proof is to show that Eq. (3) can be reduced to zero. Our actual overparametrization assumption (Definition 3) only states that the deterministic version of the model can achieve zero training loss, and this does not trivially imply that the corresponding stochastic neural network can achieve zero training loss, and this is why the result is both nontrivial and interesting. We have further clarified this point in the paragraph below Eq. (3).
>
> - In mathematical terms, let the model be written as $f=g^2  \circ g^1$, where $g^1$ is the stochastic block. Then, Def. 3 assumes that
> $$\sum_i( g^2 \circ \mathbb{E}\_{\epsilon} \circ g^1 (x_i)- y_i)^2 =0 \quad (Eq. (B)).$$
> whereas Eq. (3) reads:
> $$\sum_{i} \mathbb{E}_\epsilon(f(x_i,\epsilon)- y_i)^2 \quad (Eq. (A))$$
> What the proof achieved is this: **assuming Eq. (B), prove Eq. (A) converges to zero**.
> - Note that this is not a tautology at all. Because the second block $g^2$ is nonlinear in general, we have $ \mathbb{E}_\epsilon f(x\_i,\epsilon) \neq g^2 \circ \mathbb{E}\_{\epsilon} \circ g^1 (x_i);$
> therefore, without any further assumption on the function $f$,
> $$Eq. (B) \neq \sum_i \mathbb{E}_\epsilon (f(x_i,\epsilon) - y_i)^2,$$
> and, therefore, our result is not trivial.

---

### Decision · Action_Editors · 2023-04-20

**Recommendation:** Reject

**Comment:**

The paper's contribution will likely have a limited audience to the limited practicality and lack of sufficient demonstration of its theoretical or practical implications. Based on this, I recommend rejection at this stage. Thank you for considering this journal. I hope the comments will be helpful in improving your work. I also apologize for the extended review time. In case all the major comments are addressed, the paper is welcomed for resubmission.

**Audience:**

The paper might be of limited interest to the wider audience due to the following concerns shared by all reviewers. The key concern is that the main result depends on the assumption that training finds a specific type of minimum with the "two-layer" structure, where the second layer makes multiple copies of the first layer, hence "canceling out" the variance. However, there is no clear demonstration (theoretical nor empirical) that training does indeed find such a minimum. A closely related concern is that there is little discussion on theoretical or empirical implications of the results.

**Claims And Evidence:**

The paper studies the issue of vanishing predictive variance in stochastic neural networks. The authors prove a theorem stating that, for sufficiently wide networks, there exists a global minimum where the predictive variance of the network goes to zero. This is demonstrated by constructing a solution with vanishing variance for a two-layer network. The authors also show empirical studies further supporting the conclusions.